# The Drug Susceptibility of Non-Tuberculous Mycobacteria (NTM) in a Referral Hospital in Rome from 2018 to 2023

**DOI:** 10.3390/microorganisms12081615

**Published:** 2024-08-08

**Authors:** Antonio Mazzarelli, Carla Nisii, Angela Cannas, Antonella Vulcano, Barbara Bartolini, Federica Turchi, Ornella Butera, Alberto Rossi, Chiara De Giuli, Chiara Massimino, Chiara Stellitano, Valentina Antonelli, Ivano Petriccione, Enrico Girardi, Gina Gualano, Fabrizio Palmieri, Carla Fontana

**Affiliations:** 1Laboratory of Microbiology and Biorepository, National Institute for Infectious Diseases, INMI “Lazzaro Spallanzani”, IRCCS, Via Portuense 292, 00149 Rome, Italy; antonio.mazzarelli@inmi.it (A.M.); angela.cannas@inmi.it (A.C.); antonella.vulcano@inmi.it (A.V.); barbara.bartolini@inmi.it (B.B.); federica.turchi@inmi.it (F.T.); ornella.butera@inmi.it (O.B.); alberto.rossi@inmi.it (A.R.); chiara.degiuli@inmi.it (C.D.G.); chiara.massimino@inmi.it (C.M.) chiara.stellitano@inmi.it (C.S.); valentina.antonelli@inmi.it (V.A.); ivano.petriccione@inmi.it (I.P.); carla.fontana@inmi.it (C.F.); 2Scientific Direction, National Institute for Infectious Diseases, INMI “Lazzaro Spallanzani”, IRCCS, 00149 Rome, Italy; enrico.girardi@inmi.it; 3Respiratory Infectious Diseases Unit, National Institute for Infectious Diseases, INMI “Lazzaro Spallanzani”, IRCCS, 00149 Rome, Italy; gina.gualano@inmi.it (G.G.); fabrizio.palmieri@inmi.it (F.P.)

**Keywords:** non-tuberculous mycobacteria, diagnosis, drug resistance

## Abstract

*Background:* The treatment of non-tuberculous mycobacterial (NTM) infections is challenging because of the difficulty in obtaining phenotypic (pDST) and/or molecular (mDST) drug susceptibility testing and the need of a multi-drug regimen. *Objectives*: The objective was to describe the in vitro susceptibility patterns of various NTM species through an analysis of susceptibility results obtained on isolates collected between 2018 and 2023. *Methods:* Species identification and mutations in *rrs* or *rrl* genes (mDST) were identified by a line probe assay, while the pDST was performed by broth microdilution and interpreted according to CLSI criteria. *Results:* We analysed 337 isolates of NTM belonging to 15 species/subspecies. The *Mycobacterium avium* complex (MAC) was the most common (62%); other species identified included *M. gordonae* (11%), *M. kansasii* (5%), the *M. abscessus* complex (8%), *M. chelonae* (6%), and *M. fortuitum* (2%). The results of pDST (claritromycin and amikacin) and mDST (*rrl* and *rrs* genes) on 66 NTM strains showed that while wild-type *rrl* and *rrs* occurred in 86.3% and 94% strains, respectively, the pDST showed 88% sensitivity for clarithromycin and 57.5% for amikacin. The main incongruity was observed for macrolides. *Conclusions*: Most NTM are likely to be susceptible to macrolides and aminoglycosides. The molecular identification of resistant genotypes is accurate and strongly recommended for optimal patient management.

## 1. Introduction

The term non-tuberculous mycobacteria (NTM) encompasses a large number (>200) of *Mycobacterium* species that are mainly environmental opportunistic pathogens causing pulmonary and extrapulmonary infections in adults and cervical lymphadenitis in children [1,2,3]. NTM are classified mainly into two groups: rapidly growing mycobacteria (R-NTM), which form visible, mature colonies on solid media within 7 days, and slowly growing mycobacteria (S-NTM), which grow on solid media after 7 days.

Not all the NTM species are known to cause infections in humans, but there is growing awareness of the emergence of NTM as causative agents of infection worldwide [4,5,6,7]. In 2020, about 220,000 NTM cases were estimated in the USA, with an increase of 8.2% each year [8]. The cause of infection is likely multifactorial and behavioural (e.g., frequent exposure to water at public baths), but no single common genetic or immunological defect has been identified in these patients. It is likely that multiple pathways contribute towards host susceptibility to NTM infection, although some conditions are more strongly associated with the development of disease (e.g., genetic risk factors, including cystic fibrosis, biological therapies against tumour necrosis factor (TNF)-α, or the growing incidence of chronic lung diseases) [9,10,11]. The clinical course of NTM infection is heterogeneous, with some patients remaining stable without the need for treatment and others developing refractory diseases associated with considerable morbidity and mortality [10].

Although the improved laboratory detection assays available today, coupled with an increased awareness by physicians, results in NTM strains being detected more accurately and rapidly in clinical samples than before [12], the diagnosis of NTM disease remains laborious and often challenging. The inherent nature of NTM as environmental microorganisms introduces the potential for their presence in biological samples due to contamination or colonisation rather than true infection. For this reason, guidelines for NTM diseases establish three main criteria for diagnosis (clinical, radiological, and microbiological [13]) and provide evidence-based therapies for the most common NTM species, such as the *Mycobacterium avium* complex (MAC) and *Mycobacterium abscessus* complex [13]. No shared guidelines exist for establishing diagnostic criteria in cases of disseminated disease or NTM infection affecting sites other than the lung (e.g., skin, bones, blood, and lymph nodes) [1].

Treatment regimens are based on species identification, a phenotypic drug susceptibility test (pDST) or molecular susceptibility test (mDST), and the severity of the disease [9].

The recommended method for a pDST, and for guiding clinicians in choosing the best therapy for patients with NTM disease, it is the broth microdilution susceptibility test [14], which is carried out on viable isolates grown on solid culture. This means that weeks may be necessary to obtain a result, and in some circumstances (in the case of a strain that fails to grow), it may not even be attainable [15]. Because treatment is sometimes ineffective and frequently poorly tolerated, several alternative antibiotics are tested in association with the first- and second-line antibiotics [16]. The molecular DST (mDST) offers the advantage of species identification as well as the rapid detection of drug resistance for macrolides and aminoglycosides through genetic mutations on *rrl* and *rrs* genes, respectively [17].

In this paper, by using a combined phenotypic and molecular approach, we analysed the in vitro drug susceptibility data of slowly growing (S-NTM) and rapidly growing (R-NTM) non-tuberculous mycobacteria strains collected from 2018 to 2023.

## 2. Methods

### 2.1. Study Setting

This study was carried out at the ‘L. Spallanzani’ National Institute for infectious diseases, which is a tertiary care hospital in Rome. The study involved 337 clinical NTM isolates obtained on solid or liquid culture from January 2018 to December 2023 from patients who fulfilled the diagnostic criteria for NTM disease. The primary data extracted for each isolate included anonymised sample ID, age, sex, type of sample, sample type, identified (sub-) species of mycobacteria, and method for species identification. When antibiotic susceptibility testing was performed, susceptibility profiles and testing methods were included in the analysis.

Species identification of NTM isolates was retrieved from our Laboratory Information System (LIS). If multiple samples were available for the same patient, only the first was included in the analysis unless a different species was identified. Clinical history and treatment data were not available.

### 2.2. Molecular Identification and Molecular Resistance of NTM Strains (mDST)

GenoType NTM-DR (NTM-DR; Hain Lifescience, Nehren, Germany) is a line probe assay (LPA) that enables species- or subspecies-level identification of the major clinically encountered NTM, including MAC species (*M. avium*, *M. intracellulare*, and *M. chimera*), *M. chelonae*, and subspecies belonging to *M. abscessus*, i.e., *M. abscessus* subsp. *abscessus*, *M. abscessus* subsp. *massiliense*, and *M. abscessus* subsp. *bolletii*. The NTM-DR assay also allows for detection of antibiotic resistance to macrolides and aminoglycosides. Specifically, macrolide resistance is identified by polymorphisms (T28 or C28) at position 28 in the *erm*(41) gene and mutations at positions 2058/2059 in the *rrl* gene. Conversely, aminoglycoside resistance was determined by examining positions 1406 to 1408 of the *rrs* gene. The mutation probes for the NTM-DR assay were designed to hybridise to alleles containing specific mutations: four mutations in the *rrl* gene, including A2058C (MUT1 probe), A2058G (MUT2), A2059C (MUT3), and A2059G (MUT4), and one mutation in the *rrs* gene (A1408G). The frequencies of specific *rrs* and *rrl* mutations were evaluated on MAC and *M. abscessus* complex isolates, as per manufacturer’s instructions.

### 2.3. Phenotypic Drug Susceptibility Tests (pDSTs)

For S-NTM strains, broth microdilution assays were performed using the SLOMYCO assay (Thermo Fisher Scientific Inc., Waltham, MA, USA) on isolates grown on solid culture medium (Liofilchem, Roseto degli Abruzzi, Italy). The SLOMYCO panel contains the following drugs: amikacin, clarithromycin, ciprofloxacin, doxycycline, ethambutol, ethionamide, isoniazid, linezolid, moxifloxacin, rifabutin, rifampicin, streptomycin, and trimethoprim/sulfamethoxazole. For R-NTM, phenotypic antimicrobial resistance was tested by the RAPMYCOI assay (Thermo Fisher Scientific). The following antibiotics were tested: cotrimoxazole, ciprofloxacin, moxifloxacin, cefoxitin, cefepime, ceftriaxone, amikacin, doxycycline, tigecycline, clarithromycin, linezolid, mynocil, amoxicycline clavulanic acid, imipenem, and tobramycin.

SLOMYCO assay and RAPMYCOI assay were performed following the manufacturer’s instructions. In brief, to standardise the inoculum, a 0.5 McFarland suspension was obtained by use of a nephelometer and diluted in sterile water in cation-adjusted Mueller–Hinton broth with OADC (Oleic Acid-albumin-Dextrose-Catalase) and cation-adjusted Mueller–Hinton broth with W/TES (buffer), respectively. These suspensions (corresponding to 5 × 10^5^ cfu/mL; range 1 × 10^5^ to 1 × 10^6^ cfu/mL) were used to inoculate 96-well plates containing the lyophilised antimicrobials. Time and temperature of incubation were those suggested by the manufacturer, as most R-NTM grow best at 30° ± 2 °C while S-NTM grow more optimally at 36 ± 2 °C. The minimum inhibitory concentrations (MIC) for each drug were read at 7–14 days for S-NTM while for R-NTM, MICs were determined after 3 days of incubation, except for clarithromycin, in which the incubation period was extended to 14 days (for *M. abscessus* complex) and interpreted in compliance with the CLSI document M24 (third edition) [13].

## 3. Results

From January 2018 to December 2023, 337 isolates of NTM were identified by GenoType NTM-DR (Figure 1A).

Of the 15 species/subspecies of NTM identified, the *Mycobacterium avium* complex (MAC) was found to be the most common (208/337, 62%): 81/208 were *M. avium* subsp. *avium* (39%), 29/208 were *M. avium* subsp. *chimaera* (14%), and 23/208 were *M. intracellulare* subsp. *intracellulare* (11%). For 75/208 strains (36%), subspecies identification was not possible with the diagnostic assays available at the time of testing; these strains were classified as *M. intracellulare/chimaera*. Other S-NTM identified were *M. gordonae* (36/337, 11%) and *M. kansasii* (17/337, 5%). Rapid-growing NTM species identified were the *M. abscessus* complex (26/337, 8%), *M. chelonae* (21/337, 6%), and *M. fortuitum* (8/337, 2%). Within the *M. abscessus* complex, we identified eleven *M. abscessus* subsp. *abscessus* (11/26, 42%), two *M. abscessus* subsp. *bolletti* (2/26, 8%), and one *M. abscessus* subsp. *massiliense* (1/26, 4%). Other NTM species identified (that are less frequently isolated in clinical practice) were *M. xenopi* (8/337, 2%), *M. mucogenicum* (3/337, 1%), *M. asiaticum* (2/337, 1%), *M. malmoense* (4/337, 1%), *M. scrofulaceum* (3/337, 1%), and *M. szulgai* (1/337, 0.3%).

Samples received were mainly sputum, broncho alveolar lavage, and biopsies and blood cultures (Figure 1B).

Thirty-two isolates are classified as ‘unspecified’ because the laboratory received the strains (grown on liquid or solid media) from other hospitals for susceptibility testing, and no information on the original sample was provided.

Demographic characteristics showed that about half, 154/289 (53%), of NTM strains were isolated from female patients falling in the following age brackets: 29% were aged 26–45, 48% were 46–65, and 47.6% were >65 years (Table 1).

Male patients were 135/289 (47%), and the most represented age group was 46–65 years (54.6%), followed by the older population (>65 years, 42%), and the younger group (age 26–45, 26%). In 48 cases, demographic data are missing because the samples were received from other hospitals and no information was available.

The pDST was performed on 140/337 strains (41%), i.e., on patients that met the clinical/diagnostic criteria for NTM infection; in the absence of clinical information, for NTM species known for being more frequently associated with infection; or in cases of samples from the lower respiratory tract (Table 2).

Although the pDST panel provides a large number of antibiotics (Appendix A), the analysis was carried out only for the antibiotics for which interpretation criteria are available. The MICs of clarithromycin (CLR), amikacin (AMK), moxifloxacin (MXF), and linezolid (LZD) were interpreted in terms of S, I, or R categories. The MICs for *M. kansasii* were interpreted also for rifampicin (RIF), rifabutin (RFB), ciprofloxacin (CIP), doxycycline (DOX), and trimethoprim-sulfamethoxazole (SXT). Most species of NTM in our study were susceptible to CLR: Table 2 shows susceptibility percentages ranging from 81% for *M. intracellulare* to 93% for *M. avium* and 100% for *M. chimaera* (Table 2). *M. fortuitum* showed a lower rate of susceptibility (50%) but from only two isolates, so this figure may not be significant. MXF showed lower levels of activity, with considerable percentages of strains falling in the ‘Intermediate’ category, as high as 37% and 40% for *M. avium* and *M. chimaera*, respectively. Of note, the *M. abscessus* complex (a rapidly growing species frequently grown from children and cystic fibrosis patients) was mostly resistant (83%) to MXF. AMK also showed high levels of susceptibility, while the data obtained for LZD varied greatly, even within the MAC group (≥50% resistant strains for *M. avium* and *M. chimaera* and 28% for *M. intracellulare*) (Table 2).

The mPDT has been available at our laboratory since 2021, and the test was carried out on 125 strains. A mutation on the *rrl* gene conferring resistance to macrolides was detected in nine strains (9/125, 7.2%), while resistance to aminoglycosides (through a mutation on the *rrs* gene) was found in four strains (4/125, 3.2%). Because *M. abscessus* subsp. *abscessus* and *M. abscessus* subsp. *bolletii* could have an *erm*-inducible gene (*erm*(*41*)) coding for a macrolide-inducible ribosomal methylase, the exposure to a macrolide in such cases could lead to drug inactivation (inducible resistance). No such strains were detected in our study.

For 66 strains, both pDST and mDST data were available, and we were able to compare the susceptibility data for clarithromycin and amikacin on this group of isolates, which comprised rapid and slow growers (Table 3).

As shown in Table 3, wild-type *rrl* occurred in 57/66 (86.3%) of strains, and this result correlated well, with a 56% susceptibility found with the pDST (only 1/66 strains was found to be resistant in the pDST, although with a WT rrl genotype). The same table shows that 7/66 (10.6%) isolates gave discordant results, having been classified as resistant to clarithromycin in the mDST but showing a susceptible phenotype. Discordant results were also found for amikacin (*rrs* gene): 62/66 WT *rrs* strains vs. 38/66 susceptible in the pDST, with a considerable number of ‘intermediate’ results (16/66), and 8/66 resistant strains.

## 4. Discussion

Despite the fact that NTM are frequent colonisers of the respiratory tract as well as other body sites, the occurrence of NTM disease is increasing worldwide to the point that it could become a major public health problem in the future [18]. In our study, the isolation of NTM increased considerably over time (except during the SARS-CoV-2 outbreak), and this is in agreement with other studies [6,7,8] (Appendix A). Species belonging to the MAC group together with the *M. abscessus* complex represent the most frequently identified species, 62% and 8% [5].

Because of the difficulty in correlating a positive culture and clinical disease, patients suspected of having an NTM infection are required to meet all clinical and microbiological criteria [12,13]. For these patients, drug susceptibility data are essential for clinical management.

The Clinical and Laboratory Standards Institute (CLSI) recommends using broth microdilution in cation-adjusted Mueller–Hinton broth (CAMHB) for the pDST of most NTM [4,19,20]. There is, however, limited evidence in support of the current CLSI breakpoints in terms of reference strains, MIC distributions, epidemiological cut-off values (ECOFFs), PK/PD values, and clinical outcome [21]. Our work aimed to evaluate the results obtained by phenotypic and molecular drug susceptibility (pDST and mDST) as a way to better understand the characteristics of the strains circulating in our geographical area, especially in terms of susceptibility and resistance to the currently used antibiotics.

Although international guidelines are available, the treatment of NTM disease is mostly empirical and not entirely successful because NTM are intrinsically resistant to several drugs; therefore, patients need to be treated with combinations of antibiotics based on susceptibility testing [22].

While the diagnostic means available for *M. tuberculosis* (MTB), including molecular diagnosis and drug-susceptibility testing, are well established and standardised and possess a higher rate of sensitivity and specificity, the assays available for NTM infections are not well standardised. In general, drug-sensitive MTB is effectively treated with a standard multi-drug regimen containing well-defined first- and second-line antibiotics. In contrast, NTM species display significant heterogeneity in their susceptibility to standard drugs effective on other mycobacteria, such as MTB [23].

Our data, obtained by both a pDST and mDST, showed that macrolides and aminoglycosides were the most effective drugs against NTM. CLR is a type of macrolide antibiotic that works by binding to the large ribosomal subunit located near the peptidyl transferase centre. It inhibits protein synthesis, which leads to the cessation of bacterial growth. This drug class is the only one for which there is a proven correlation between in vitro susceptibility test results and in vivo clinical response [24]. CLR can be administered orally [25] and is highly active against many species of NTM both S-NTM and R-NTM. Our pDST data confirmed that CLR was the most effective drug for the MAC and for the other non-MAC NTM. In particular, *M. avium* subsp. *avium* and *M. avium* subsp. *chimaera* represent the two most sensitive species of the MAC group (Table 2). Conversely, *M. intracellulare* was the species with the highest resistance rates.

Although ethambutol, rifampin, rifabutin, and streptomycin are useful clinically, breakpoints for determining susceptibility and resistance have not been established, and previous studies show that there is no correlation between in vitro MIC results and clinical response in patients with the MAC [26].

Our data showed that AMK is the second most effective drug (Table 2). It is an aminoglycoside that binds to bacterial 30S ribosomal subunits and leads to the disruption of normal protein synthesis and the production of nonfunctional or toxic peptides. AMK is also considered an effective first-line drug for the treatment of NTM [27,28].

As the second-largest NTM species and one of the most notorious causative agents of disease, the drug therapy of *M. abscessus* complex disease is long and resistant to several antibiotics, and it is often associated with drug-related adverse effects, leading to ineffective treatment results [29]. The combination therapy of intravenous amikacin with cefoxitin or imipenem and an oral macrolide has been recommended by the ATS/IDSA and many other experts [24]. However, treatment response rates are not excellent, and optimal therapeutic regimens and treatment durations are not well established. Two new *M. abscessus* complex-related species, *M. abscessus* subsp. *massiliense* and *M. abscessus* subsp. *bolletii*, can be discerned thanks to improvements in microbiological and molecular techniques, but the data regarding in vitro DST results of these new subspecies are limited [30]. It has been reported that for *M. abscessus* subsp. *abscessus* and *M. abscessus* subsp. *bolletii*, there is a poor correlation between DST results in vitro and the treatment response in vivo [30].

The detection of acquired resistance is fundamental before starting treatment or at a recurrence of NTM disease. However, the time needed to perform a pDST for detecting any resistance can be as long as 6 weeks for the MAC and 14 days for the *M. abscessus* complex [31]. Molecular DST techniques are useful in that they allow for the early detection of mutations conferring resistance, even on the clinical sample.

In this regard, our study showed that just 10.6% of our strains showed a mutation on the *rrl* gene in the mDST. It is well known that acquired CLR resistance is associated with mutations of the *rrl* gene at positions 2058/2059 [32]. Indeed, these point mutations are present at a high frequency in clarithromycin-resistant strains [33,34]. In addition, inducible resistance to CLR has been reported in *M. abscessus* subsp. *abscessus* and *M. abscessus* subsp. *bolletii* due to the induced synthesis of an RNA methylase encoded by the *erm*(41) gene [35]. Macrolide resistance is identified by polymorphisms (T28 or C28) at position 28 in the *erm*(41) gene and mutations at positions 2058/2059 in the *rrl* gene; *M. abscessus* subsp. *abscessus* and *M. abscessus* subsp. *bolletii* may exhibit such T/C polymorphism [36]. Furthermore, inducible resistance does not occur in *M. abscessus* subsp. *massiliense* because it has a partially deleted, nonfunctional *erm*(41) gene [35].

Mutations in the *rrs* gene responsible for AMK resistance occur mainly at position 1408 in both the MAC and *M. abscessus* complex [36]. In our study, among four isolates that were resistant to AMK (6%), all harboured the A1408G mutation, and no strains showed inducible resistance.

However, we observed discrepancies between pDST and mDST results; all seven strains carrying mutations conferring resistance to macrolides (*rrl* gene), were susceptible in the pDST. Conversely, eight strains with a wild-type *rrs* gene were phenotypically resistant. One hypothesis to explain this discrepancy is that both DST assays were performed on heterogeneous populations composed of strains with a wild-type and a resistance-conferring allele. This is in agreement with the work of Mougari et al. who reported that a heterogeneous population containing a resistance mutation that is not screened for by a specific probe can be missed if a wild-type population is also present [37]. The second hypothesis is that the discrepancies showed are probably due to the low standardisation and reproducibility of the pDST [38].

Some limitations of this study need to be mentioned. First, although most of the strains and/or patients with an NTM disease in the Lazio region are referred to our hospital, the study used data from a single centre, and this could explain why, for example, we observed a minority of R-NTM compared to S-NTM. Second, because of the difficulty in obtaining clinical data, this study could not distinguish isolates from treatment-naive versus treatment-experienced patients. Despite these limitations, the relatively large collection of strains studied provides useful information on the performance and concordance or discordance of the DST assays currently available.

Hopefully, the use of new, rapid, and standardised phenotypic and genotypic tests will in the near future allow for a better definition of drug susceptibility, at least for some NTM species. To overcome the long turnaround time of culture results for mycobacteria, the role of improved molecular methods for the genotyping of NTM and the identification of resistance determinants will be key to choosing the best treatment regimens.

## Figures and Tables

**Figure 1 microorganisms-12-01615-f001:**
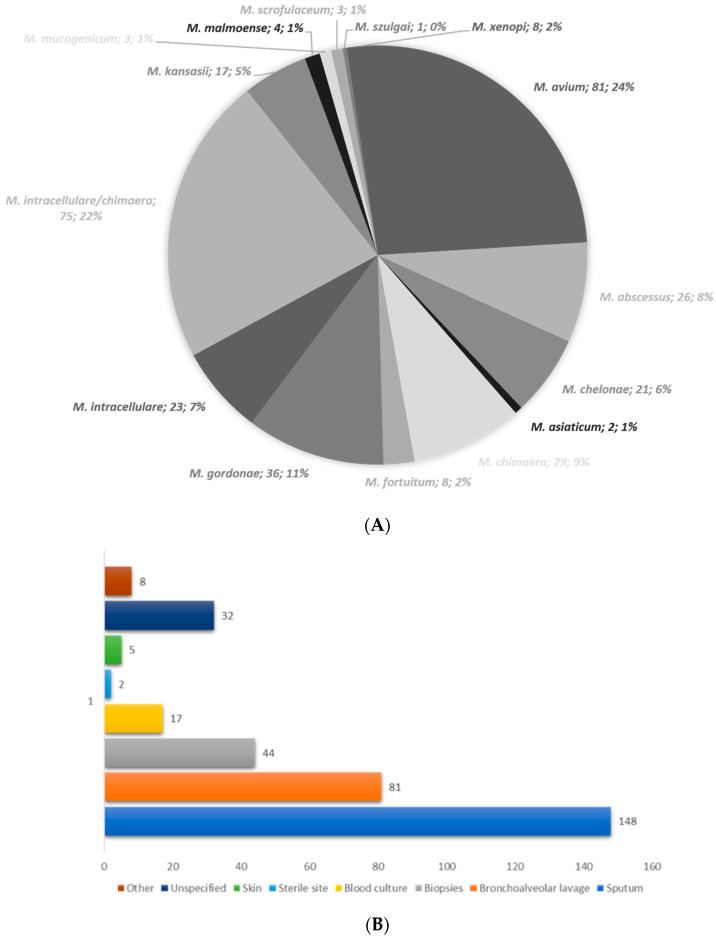
(**A**) Species distribution among the 337 isolates of NTM identified as belonging to fifteen different species/subspecies. (**B**) The distribution of the most frequent specimen types received.

**Table 1 microorganisms-12-01615-t001:** Demographic characteristics of NTM patients.

	N°	Female				N°	Male			
		Age Groups %					Age Groups %			
		<25	26–45	46–65	>65		<25	26–45	46–65	>65
*M. avium*	41	-	5	29	53	30	-	26	40	33
*M. abscessus* complex	13	-	23	23	57	10	-	20	30	50
*M. chelonae*	11	-	27	36	27	9	-	22	44	33
*M. asiaticum*	1	-	-	100	-	1	-	-	100	-
*M. chimaera*	9	-	22	-	77	10	-	40	30	30
*M. fortuitum*	2	-	50	50	-	4	-	-	100	-
*M. gordonae*	16	6	25	43	25	16	6	25	31	37
*M. intracellulare*	10	-	10	40	50	5	-	-	40	60
*M. intracellulare/chimaera*	32	-	6	37	53	37	-	2	24	68
*M. kansasii*	9	-	33	44	22	4	25	50	-	25
*M. malmoense*	1	-	-	100	-	3	-	-	67	33
*M. mucogenicum*	2	-	50	-	50	1	100	-	-	-
*M. scrofulaceum*	2	-	-	-	100	1	-	-	100	-
*M. szulgai*	1	-	-	-	100	-	-	-	-	-
*M. xenopi*	4	-	75	25	-	4	-	-	50	50
Total	154		29	48	47.6	135		26	54.6	42

**Table 2 microorganisms-12-01615-t002:** pDST for slowly growing non-tuberculous mycobacteria and rapidly growing non-tuberculous mycobacteria isolates from 2018 to 2023.

Organism	N° Strain	Parameters	CLR	RFB	MXF	RIF	SXT	AMK	LZD	CIP	DOX
*M. avium*	60	Susceptible (%)	93	-	34	-	-	72	22	-	-
		Intermediate (%)	5	-	37	-	-	15	20	-	-
		Resistant (%)	1	-	27	-	-	3	58	-	-
*M. intracellulare*	23	Susceptible (%)	81	-	38	-	-	67	38	-	-
		Intermediate (%)	-	-	28	-	-	14	33	-	-
		Resistant (%)	19	-	28	-	-	14	28	-	-
*M. chelonae*	2	Susceptible (%)	100	-	100	-	-	100	100	100	100
		Intermediate (%)	-	-	-	-	-	-	-	-	-
		Resistant (%)	-	-	-	-	100	-	-	-	-
*M. abscessus* complex	12	Susceptible (%)	100	-	8	-	16	75	58	16	-
		Intermediate (%)	-	-	8	-	-	8	33	25	25
		Resistant (%)	-	-	83	-	83	16	8	58	75
*M. chimaera*	30	Susceptible (%)	100	-	10	-	-	63	16	-	-
		Intermediate (%)	-	-	40	-	-	23	36	-	-
		Resistant (%)	-	-	46	-	-	13	50	-	-
*M. fortuitum*	2	Susceptible (%)	50	-	100	-	100	100	100	-	100
		Intermediate (%)	-	-	-	-	-	-	-	-	-
		Resistant (%)	50	-	-	-	-	-	-	100	-
*M. gordonae*	2	Susceptible (%)	100	100	100	50	50	100	100	50	-
		Intermediate (%)	-	-	-	-	-	-	-	50	100
		Resistant (%)	-	-	-	50	50	-	-	-	-
*M. kansasii*	9	Susceptible (%)	100	78	67	22	11	67	44	11	-
		Intermediate (%)	-	-	33	-	-	22	22	44	44
		Resistant (%)	-	22	-	78	89	11	22	44	55

A dash (-) indicates the absence of CLSI breakpoints. AMK, amikacin; CIP, ciprofloxacin; CLR, clarithromycin; DOX, doxycycline; LZD, linezolid; MXF, moxifloxacin; RFB, rifabutin; RIF, rifampicin; SXT, trimethoprim/sulfamethoxazole. AMK’s MIC breakpoints: S, ≤16 μg/mL; I, 32 μg/mL; R, ≥64 μg/mL. CLR’s MIC breakpoints: S, ≤8 μg/mL; I, 16 μg/mL; R, ≥32 μg/mL. MXF breakpoints S ≤ 1 μg/mL; I, 2 μg/mL; R, ≥4 μg/mL; LZD breakpoints S ≤ 8 μg/mL; I, 16 μg/mL; R, ≥32 μg/mL. Interpretations based on [13].

**Table 3 microorganisms-12-01615-t003:** Comparison between *rrs* and *rrl* genotypes (mDST) and susceptibility phenotypes (Claritromycin (CLR) and Amikacin (AMK)). (pDST) on 66 NTM isolates for which pDST and mDST data were available. Strains were included regardless of species identification.

	Molecular DST (N°, %)	pDST, CLR/AMK (N°, %)		
		S	I	R
WT *rrl* (CLR-S)	57/66 (86.3)	56/66 (84.8)	-	1/66 (1.5)
Mutant *rrl* (CLR-r)	7/66 (10.6)	7/66 (10.6)	-	-
WT *rrs* (AMK-S)	62/66 (94)	38/66 (57.5)	16/66 (24.2)	8/66 (12)
Mutant *rrs* (AMK-R)	4/66 (6)	1/66 (1.5)	2/66 (3)	1/66 (1.5)

## Data Availability

Data can be found in the ad hoc created Excel database, which will be archived at the authors’ institution (INMI L. Spallanzani IRCCS, Rome, Italy).

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
