# Peer review of "The Drug Susceptibility of Non-Tuberculous Mycobacteria (NTM) in a Referral Hospital in Rome from 2018 to 2023"

_microorganisms, 2024, doi:10.3390/microorganisms12081615_

Round 1

Reviewer 1 Report

Comments and Suggestions for Authors

Major comment

The authors have performed phenotypic susceptibility testing of 140 isolates of NTM, of which 16 were rapidly growing strains.

In the CLSI M24 guidelines (section 5.2.1), it is recommended that rapidly growing NTM are tested using cation-adjusted Mueller-Hinton broth and that cation-adjusted Mueller-Hinton broth with AODC supplement is used for slow-growing NTM.

The authors appear to have applied the Thermofisher panel for slow growing mycobacteria (SLOMYCO assay) for the testing of rapidly growing strains. They have also used MH broth + OADC for the testing of rapidly growing strains, whereas MH broth (without OADC) is recommended. Thermofisher offer a separate panel and a separate broth dedicated to the testing of rapidly growing NTM (the Sensititre RAPMYCOI AST Plate).

This raises three concerns for me. The first is that I do not know how the presence of presence of AODC will influence the MIC results for rapidly growing NTM. Secondly, the use of the correct panel would have allowed a more appropriate range of antibiotics to be tested for rapidly-growing NTM. Finally and most importantly, the authors state that MIC readings for rapidly growing strains were performed after 72 h (line 107) but it is not possible to detect inducible resistance to clarithromycin without incubation for at least 14 days in accordance with CLSI M24; section 5.8.4. (and also the instructions for use for the Sensititre RAPMYCOI AST Plate). It is notable that no inducible resistance was detected in the 12 strains of M. abscessus complex tested in this paper, yet such resistance is prevalent in this group.

Only a minority of the data presented here are for rapidly-growing NTM. The authors are invited to defend their choice of methods and/or, if appropriate, add a further paragraph to the discussion section to highlight further limitations of their study.

Other comments:

Abstract

Line 20: Apply italics to the words Mycobacterium avium.

Lines 20 – 22: Why do the percentages quoted here not match the figures shown in Fig 1A? For example, the occurrence of M. fortuitum is stated to be 0.2% (in fig 1A it is quoted as 2% - 10 x higher), also M. abscessus (7% versus 8%), Please check the numbers for every species and ensure only the most prevalent are quoted in the abstract.

Introduction:

Line 35: NTM stands for non-tuberculous mycobacteria, which is plural. It is therefore unnecessary (and incorrect) to use the abbreviation “NTMs”. Please correct this throughout the whole paper.

Line 36: I wonder whether the term “behavioural” is appropriate. What “behavior” of an individual might predispose to NTM infection? If such examples exist (e.g. smoking), at least one example should be provided to justify the term.

Methods

Line 86: Italics are not required for “subsp.” (please change throughout the whole paper).

Results

Figure 1B:

Replace “emoculture” with “blood culture”.

It is surprising (for a published study) that almost 10% of samples were unknown/unspecified. Is it not possible to determine the sample type (even if it is necessary to contact other hospitals)?

Line 115: M. avium sensu stricto can be replaced with M. avium subsp. avium.

Lines 114 - 126: As with the abstract, there are discrepancies with the numbers. For example, in line 120 the occurrence of M. abscessus is stated to be 7% - whereas in Fig. 1A it is 8%. (see also line 197). The figure for M. fortuitum is also incorrect. Many other figures are also incorrect, e.g. 1/337 = 0.3% (not 0.2%), 2/337 = 0.6% (not 0.5%), etc… Please check the numbers for every species. Comment: since the final version of the paper was presumably checked by all 17 authors, it is disappointing that these basic errors were not detected and naturally raises concerns about other areas of the paper – all authors are encouraged to carefully re-check any revised version.

Line 155: “The MICs of M. kansasii……”. Replace ‘of’ with ’for’. (you can’t have a minimum inhibitory concentration of bacteria).

Line 160: “MXF showed lower levels of susceptibility”. Antibiotics cannot show “susceptibility” – only activity. (see also line 164).

Line 174: It is surprising that none of 12 isolates of M. abscessus showed inducible resistance to claithromycin. M. abscessus subsp. abscessus typically shows such resistance (in contrast to M. abscessus subsp. massiliense). Since no sub-speciation has been performed it could be misleading to refer simply to “M. abscessus” – especially when quoting susceptibility data. I recommend changing “M. abscessus” to M. abscessus complex” throughout the whole paper, wherever it is mentioned in reference to the isolates collected in this study (including figures).

Line 188: When using S/I/R based on CLSI criteria, ‘I’ denotes ‘intermediate’ rather than ‘indeterminate’ – which are two quite different things.

Line 191: the word ‘despite’ appears to be mis-placed.

Line 193: “identification of NTMs”. Do you mean detection (or isolation) of NTM?

Line 220: no capital letter required for “mycobacteria”. (see also line 291).

Line 227: replace ‘NTN’ with NTM’

Line 231: It is inappropriate to make any generalization for M. fortuitum based on testing only 2 strains. Please delete.

Line 241: do not use italics for the word “complex”. (italics are only necessary for Latin words – not ‘complex’ or ‘subspecies’).

The reference section requires consistent formatting applied to all references. Just as one example, in the title of the article for reference 37, capitalization is used for words such as “Susceptibility” – this is in contrast to, for example, reference 34. The authors should consult the instructions to authors and/or a recent paper from Microorganisms and apply the correct formatting to each reference.

Comments on the Quality of English Language

The quality of English is generally fine. Some minor editing might be appropriate.

Author Response

Major comment

The authors have performed phenotypic susceptibility testing of 140 isolates of NTM, of which 16 were rapidly growing strains.

In the CLSI M24 guidelines (section 5.2.1), it is recommended that rapidly growing NTM are tested using cation-adjusted Mueller-Hinton broth and that cation-adjusted Mueller-Hinton broth with AODC supplement is used for slow-growing NTM.

The authors appear to have applied the Thermofisher panel for slow growing mycobacteria (SLOMYCO assay) for the testing of rapidly growing strains. They have also used MH broth + OADC for the testing of rapidly growing strains, whereas MH broth (without OADC) is recommended. Thermofisher offer a separate panel and a separate broth dedicated to the testing of rapidly growing NTM (the Sensititre RAPMYCOI AST Plate).

This raises three concerns for me. The first is that I do not know how the presence of presence of AODC will influence the MIC results for rapidly growing NTM. Secondly, the use of the correct panel would have allowed a more appropriate range of antibiotics to be tested for rapidly-growing NTM. Finally and most importantly, the authors state that MIC readings for rapidly growing strains were performed after 72 h (line 107) but it is not possible to detect inducible resistance to clarithromycin without incubation for at least 14 days in accordance with CLSI M24; section 5.8.4. (and also the instructions for use for the Sensititre RAPMYCOI AST Plate). It is notable that no inducible resistance was detected in the 12 strains of M. abscessus complex tested in this paper, yet such resistance is prevalent in this group.

Reply:

We sincerely apologize for the oversight of not specifying in the methods the use of the RAPMYCO AST Plate. We did in fact use the appropriate Sensititre assay, as testified by the newly added Supplementary Table 1 (Table S1), where the MICs for all the tested antibiotics are reported. We have amended the text (Page 5, lines 113-133).

As for the problem of the inducible resistance in rapidly growing strains, we did perform a further reading after 14 days of incubation, we did not specify it because we stated that no strains showed such resistance. But we agree with the reviewer that it is best to specify this so we amended the text (Page 6, lines 131-132).

Only a minority of the data presented here are for rapidly-growing NTM. The authors are invited to defend their choice of methods and/or, if appropriate, add a further paragraph to the discussion section to highlight further limitations of their study.

Reply:

We believe that this could be a reflection of the fact that this study is based on the experience of a single centre. We agree with the reviewer and mentioned this as another limitations of the study (Page 15, lines 312-313).

Other comments:

Abstract

Line 20: Apply italics to the words Mycobacterium avium.

Done

Lines 20 – 22: Why do the percentages quoted here not match the figures shown in Fig 1A? For example, the occurrence of M. fortuitum is stated to be 0.2% (in fig 1A it is quoted as 2% - 10 x higher), also M. abscessus (7% versus 8%), Please check the numbers for every species and ensure only the most prevalent are quoted in the abstract.

Reply:

For M. fortuitum, it was a mistake, thanks to the reviewer for pointing it out. We amended the text. For M. abscessus it was the rounding up of the figures in the construction of the pie chart that caused the problem. For homogeneity, we used the rounded up figure also in the text. (Page 7, line 148)

Introduction:

Line 35: NTM stands for non-tuberculous mycobacteria, which is plural. It is therefore unnecessary (and incorrect) to use the abbreviation “NTMs”. Please correct this throughout the whole paper.

Done throughout

Line 36: I wonder whether the term “behavioural” is appropriate. What “behavior” of an individual might predispose to NTM infection? If such examples exist (e.g. smoking), at least one example should be provided to justify the term.

Reply:

We amended the text (Page 3, line 53)

Methods

Line 86: Italics are not required for “subsp.” (please change throughout the whole paper).

Done throughout

Results

Figure 1B:

Replace “emoculture” with “blood culture”.

Done

It is surprising (for a published study) that almost 10% of samples were unknown/unspecified. Is it not possible to determine the sample type (even if it is necessary to contact other hospitals)?

Reply:

This unfortunately is a frequent problem for hospitals (like ours) that carry out research on diagnostic residual samples. If a hospital sends a strain (isolated at their laboratory) requesting a diagnostic laboratory service they are not required to provide information on the original material, as it is not necessary for the purpose of the analysis. If would be extremely complicated, if not impossible, to contact them months or even years later!

Line 115: M. avium sensu stricto can be replaced with M. avium subsp. avium.

Done

Lines 114 - 126: As with the abstract, there are discrepancies with the numbers. For example, in line 120 the occurrence of M. abscessus is stated to be 7% - whereas in Fig. 1A it is 8%. (see also line 197). The figure for M. fortuitum is also incorrect. Many other figures are also incorrect, e.g. 1/337 = 0.3% (not 0.2%), 2/337 = 0.6% (not 0.5%), etc… Please check the numbers for every species. Comment: since the final version of the paper was presumably checked by all 17 authors, it is disappointing that these basic errors were not detected and naturally raises concerns about other areas of the paper – all authors are encouraged to carefully re-check any revised version.

 Reply:

We have checked and corrected all the errors throughout the paper.

Line 155: “The MICs of M. kansasii……”. Replace ‘of’ with ’for’. (you can’t have a minimum inhibitory concentration of bacteria).

Done

Line 160: “MXF showed lower levels of susceptibility”. Antibiotics cannot show “susceptibility” – only activity. (see also line 164).

Done (Page 10, line 193)

Line 174: It is surprising that none of 12 isolates of M. abscessus showed inducible resistance to claithromycin. M. abscessus subsp. abscessus typically shows such resistance (in contrast to M. abscessus subsp. massiliense). Since no sub-speciation has been performed it could be misleading to refer simply to “M. abscessus” – especially when quoting susceptibility data. I recommend changing “M. abscessus” to M. abscessus complex” throughout the whole paper, wherever it is mentioned in reference to the isolates collected in this study (including figures).

Reply:

We corrected the text as correctly suggested by the reviewer (Page 7, lines 148-151), and in Tables 1 and 2.

Line 188: When using S/I/R based on CLSI criteria, ‘I’ denotes ‘intermediate’ rather than ‘indeterminate’ – which are two quite different things.

Reply:

The reviewer is correct, we amended the text (Page 11, line 222).

Line 191: the word ‘despite’ appears to be mis-placed.

Reply:

We amended the text to clarify our meaning (Page 12, line 228).

Line 193: “identification of NTMs”. Do you mean detection (or isolation) of NTM?

Reply:

We corrected as suggested by the reviewer (Page 12, line 228).

Line 220: no capital letter required for “mycobacteria”. (see also line 291).

Done throughout the paper

Line 227: replace ‘NTN’ with NTM’

Done (Page 13, line 259).

Line 231: It is inappropriate to make any generalization for M. fortuitum based on testing only 2 strains. Please delete.

Reply:

We amended the text (Page 13, line 262).

Line 241: do not use italics for the word “complex”. (italics are only necessary for Latin words – not ‘complex’ or ‘subspecies’).

Done throughout the paper

The reference section requires consistent formatting applied to all references. Just as one example, in the title of the article for reference 37, capitalization is used for words such as “Susceptibility” – this is in contrast to, for example, reference 34. The authors should consult the instructions to authors and/or a recent paper from Microorganisms and apply the correct formatting to each reference.

Reply:

We checked the reference list for formatting and consistency.

Reviewer 2 Report

Comments and Suggestions for Authors

Attached 

Author Response

names in Fig1A.

Reply:

we have corrected as requested throughout the manuscript.

 Line 35: “… growing awareness of the emergence of NTMs as causative agents of infection worldwide…” The authors should include information on the prevalence of the NTM. E.g.,

  1. Zhou, Y., Mu, W., Zhang, J., Wen, S.W. and Pakhale, S., 2022. Global prevalence of non-tuberculous mycobacteria in adults with non-cystic fibrosis bronchiectasis 2006–2021: a systematic review and meta-analysis. BMJ open, 12(8), p.e055672.
  2. Przybylski, G., Bukowski, J., Kowalska, W., Pilaczyńska-Cemel, M. and Krawiecka, D., 2023. Trends from the Last Decade with Nontuberculous Mycobacteria Lung Disease (NTM-LD): Clinicians’ Perspectives in Regional Center of Pulmonology in Bydgoszcz, Poland. Pathogens, 12(8), p.988.
  3. Ratnatunga, C.N., Lutzky, V.P., Kupz, A., Doolan, D.L., Reid, D.W., Field, M., Bell, S.C., Thomson, R.M. and Miles, J.J., 2020. The rise of non-tuberculosis mycobacterial lung disease. Front Immunol 11: 303.

Reply:

We have amended the reference list as suggested.

Authors can also include information on projected NTM cases. These inculded information will emphasize the global emerging problem of NTM cases.

Reply:

We included some information on projected NTM cases (Page 3, line 52), and added reference  8 that provides a link to the ntminfo.org website.

 Line 104: “….and diluted in sterile water in cation-adjusted …..”Authors should mention the amount of bacteria added per well.

Reply:

We added the required information (Page 6, line 126).

 Line 146: “…..A dash (–) indicates the absence of CLSI breakpoints..” The authors can add a supplementary table of the MIC values of all the isolates and the antibiotics tested.

This information will significantly contribute to the field of mycobacteriology by providing valuable insights into the MICs for certain non-tuberculous mycobacteria (NTM). As the breakpoints are established based on a combination of clinical data, population distribution data, comparative analysis with breakpoints for other organisms, and the knowledge of mycobacteriology experts.

Reply:

This was a very good suggestion, we have added Supplementary Table 1 (Page 10, line 185).
